# Type IIa RPTPs and Glycans: Roles in Axon Regeneration and Synaptogenesis

**DOI:** 10.3390/ijms22115524

**Published:** 2021-05-24

**Authors:** Kazuma Sakamoto, Tomoya Ozaki, Yuji Suzuki, Kenji Kadomatsu

**Affiliations:** 1Department of Biochemistry, Nagoya University Graduate School of Medicine, 65 Tsurumai-cho, Showa-ku, Nagoya 466-8550, Japan; sakamoto@med.nagoya-u.ac.jp (K.S.); tozaki@med.nagoya-u.ac.jp (T.O.); yj-suzuki@med.nagoya-u.ac.jp (Y.S.); 2Institute for Glyco-core Research (iGCORE), Nagoya University, Furo-cho, Chikusa-ku, Nagoya 464-8601, Japan

**Keywords:** axon regeneration, chondroitin sulfate, heparan sulfate, LAR, PTPδ, PTPσ, synapse

## Abstract

Type IIa receptor tyrosine phosphatases (RPTPs) play pivotal roles in neuronal network formation. It is emerging that the interactions of RPTPs with glycans, i.e., chondroitin sulfate (CS) and heparan sulfate (HS), are critical for their functions. We highlight here the significance of these interactions in axon regeneration and synaptogenesis. For example, PTPσ, a member of type IIa RPTPs, on axon terminals is monomerized and activated by the extracellular CS deposited in neural injuries, dephosphorylates cortactin, disrupts autophagy flux, and consequently inhibits axon regeneration. In contrast, HS induces PTPσ oligomerization, suppresses PTPσ phosphatase activity, and promotes axon regeneration. PTPσ also serves as an organizer of excitatory synapses. PTPσ and neurexin bind one another on presynapses and further bind to postsynaptic leucine-rich repeat transmembrane protein 4 (LRRTM4). Neurexin is now known as a heparan sulfate proteoglycan (HSPG), and its HS is essential for the binding between these three molecules. Another HSPG, glypican 4, binds to presynaptic PTPσ and postsynaptic LRRTM4 in an HS-dependent manner. Type IIa RPTPs are also involved in the formation of excitatory and inhibitory synapses by heterophilic binding to a variety of postsynaptic partners. We also discuss the important issue of possible mechanisms coordinating axon extension and synapse formation.

## 1. Introduction

Type IIa receptor tyrosine phosphatases (RPTPs) have received special attention for three reasons. 1. They serve as presynaptic adhesion molecules that form synapses with specific postsynaptic partners. For example, presynaptic PTPσ, a type IIa RPTP, serves as a synaptic organizer by heterophilic binding to its postsynaptic partners, such as TrkC, Slitrks, and LRRTM4. 2. They function as receptors for glycans, i.e., heparan sulfate (HS) and chondroitin sulfate (CS), in the regulation of axon regeneration. Extracellular chondroitin sulfate proteoglycans (CSPGs) overproduced in neural injury serve as glycan ligands to activate the enzymatic activity of PTPσ, and the downstream signaling disrupts autophagy flux and consequently inhibits axon growth. In contrast, heparan sulfate proteoglycans (HSPGs) suppress PTPσ activity and promote axon growth. 3. Combined with 1 and 2, another recent advance in this field is that a glycan, i.e., HS, is now accepted as significant in synapse formation involving type IIa RPTPs.

Focusing on the three points above, this review comprehensively describes the roles of type IIa RPTPs and glycans in axon regeneration and synaptogenesis. In addition, although axon extension and synapse formation may be orchestrated during development, the underlying regulating mechanisms are still elusive. Extending axons may need to arrest before forming synapses with their target neurons, and such arrest might be regulated by the interactions between type IIa RPTPs and glycans. We also discuss this issue.

## 2. Roles of Type IIa RPTPs and Glycans in Axon Regeneration and Synaptogenesis

### 2.1. Structures of Type IIa RPTPs

The type IIa RPTP family consists of three members, LAR, PTPδ, and PTPσ. Each member contains three immunoglobulin-like (Ig) domains, 3–8 fibronectin III (FNIII) domains in the extracellular region, a single transmembrane domain, and the intracellular region composed of a catalytic domain D1 and non-catalytic domain D2 (Figure 1). Substrates for the type IIa RPTPs have not been fully identified yet. This issue will also be discussed later.

The Ig domains are essential for ligand binding. CS and HS bind to the same site of the first Ig domain, Ig1 [1]. There are four multiple-splice sites, named MeA-D (Figure 1). MeA and MeB are particularly important for binding to most postsynaptic ligands, e.g., neurotrophin receptor tyrosine kinase C (TrkC) [2], interleukin-1 receptor accessory protein-like 1 (IL1RAPL1) [3,4], interleukin-1 receptor accessory protein (IL1RAcp) [5], Slit- and Trk-like proteins (Slitrks) [6], and synaptic adhesion-like molecule 3 (SALM3) [7], as the presence or absence of inserts at these sites influences the binding affinity. Although the presynaptic HSPG glypicans (GPCs) also bind to type IIa RPTPs, these alternative splicings do not affect the bindings of GPCs and type IIa RPTPs [8]. This is probably because HS in the GPC moiety is essential for the binding, and it binds to the Ig1 domain as described above (Figure 1). This may also be the case for the cis-binding between the presynaptic HSPG neurexin and the presynaptic PTPσ. In contrast to the above cases, the FNIII domains of type IIa RPTPs are critical for the binding to another important postsynaptic ligand, netrin-G ligand 3 (NGL3) [9,10]. The biological significance of these bindings will be discussed later.

It has been proposed that cis-dimerization of the oligomerization of RPTP negatively regulates phosphatase activity through the interaction between intracellular domains (this is the so-called wedge model) (Figure 1) [11,12,13], although the relative orientation of D1 and D2 domains upon ligand-induced clustering remains to be further verified [13]. Type IIa RPTPs have a wedge-shaped helix-loop-helix located between the membrane-proximal region and the D1 catalytic domain (Figure 1) [11]. The LAR wedge TAT peptide successfully inhibits LAR function [14]. Intracellular sigma peptide (ISP), a peptide-mimetic of the PTPσ wedge with a TAT domain, can also suppress PTPσ activity [15].

### 2.2. Axon Regeneration

#### 2.2.1. Type IIa RPTPs and Glycosaminoglycans in Axon Regeneration

Neural injuries damage axons. The portion distal to the injury site undergoes so-called Wallerian degeneration and thus disappears after injury, whereas the proximal portion of the axon regenerates if the injury happens in the peripheral nervous system. However, axons hardly regenerate in the central nervous system (CNS). This is because of the low intrinsic capacity of regeneration of the CNS neurons and inhibitors emerging upon injury. Injuries activate astrocytes and microglia and destroy the myelin sheath. These produce a variety of inhibitors of axon regeneration, which can be categorized into three classes [16,17]: 1. myelin-derived inhibitors Nogo, MAG, and OMgp; 2. canonical guidance molecules semaphorins (Sema), ephrins, and slits; and 3. glycosaminoglycans (GAGs): CS and keratan sulfate (KS).

GAGs are long sugar chains (glycans) composed of repeating disaccharide units. Five members, i.e., CS, KS, HS, dermatan sulfate, and hyaluronic acid, belong to GAGs (Figure 2). With the exception of hyaluronic acid, all the members are covalently attached to core proteins. Such glycoproteins are called proteoglycans. For example, a CS-, KS-, or HS-carrying proteoglycan is called CSPG, keratan sulfate proteoglycan (KSPG), or HSPG, respectively. It is well known that CSPGs regulate neuronal activities [18]. CSPGs downregulate neurite outgrowth of various neurons, such as dorsal root ganglion and cerebellar granular neurons [19,20,21]. It was demonstrated in 1990 that E9 chick dorsal root ganglion (DRG) axons could not pass through a strip of CS/KSPGs [22]. Then, it was shown in 2001 that the CS-digesting enzyme chondroitinase ABC could enhance axon regeneration of the nigrostriatal tract after rat brain injury [23], and in 2002, that chondroitinase ABC promoted axon regeneration and functional recovery after spinal cord injury in rats [24]. On the other hand, KS-deficient mice showed elevated neurite growth after a cortical stab [25], and better axon regeneration and functional recovery after spinal cord injury [26]. The KS-digesting enzyme keratanase II could promote axon regeneration and functional recovery after spinal cord injury in rats [27]. Therefore, CS and KS are potent inhibitors of axon regeneration. Indeed, CS and KS are extracellularly deposited in the injury area in the CNS.

CSPGs also regulate synaptic plasticity by suppressing the lateral diffusion of AMPA receptors, which is important for learning and memory [28]. Enzymatic digestion of CS stimulates dendritic spine formation [29]. These reports represent that CS works as an inhibitory molecule for synaptic plasticity. CSPGs are also involved in experience-dependent plasticity in the brain. Thus, CSPGs form a lattice-like structure, so-called perineuronal nets, in parvalbumin-positive interneurons. The perineuronal nets are formed in the critical period during development and suppress experience-dependent plasticity in adults. Chondroitinase ABC can digest CS and break perineuronal nets in adults. Administration of chondroitinase ABC into the visual cortex of monocular deprived rats or amygdala of rats with fear memory recovers ocular dominance plasticity or erasure of fear memory, respectively [30,31]. This plasticity is closely related to sulfation patterns of CS. CS6ST-1, an enzyme that sulfates the C-6 position of GalNAc in CS, produces 6-sulfated CS in infants. As animals grow, C-6 sulfation decreases and C-4 sulfation becomes dominant in the adult. Adult C6ST-1 transgenic mice strongly express 6-sulfated CS and can gain ocular dominance plasticity. In fact, the structure of the perineuronal net is loose even in adulthood in these mice [32]. Moreover, CSPGs are implicated in neurodegenerative diseases, such as Alzheimer’s disease, Parkinson’s disease, and amyotrophic lateral sclerosis [33].

Nearly a century ago, Santiago Ramón y Cajal drew injured axon terminals, which swell and form so-called dystrophic endballs (also referred to as sterile clubs or dystrophic end bulbs) [34]. It is known that dystrophic endballs stay where they are formed for several decades after CNS injuries in humans [35]. This phenomenon can be reproduced in vitro if rat adult DRG neurons are cultured on a substratum of CS gradient [19]. With this method, a spot of a mixture of laminin and the CSPG aggrecan is dried, and the rim of the spot forms a gradient of CS (aggrecan). If adult DRG neurons are cultured in the spot, the axons of DRG neurons will extend outward but arrest at the rim and form dystrophic endballs. This spot assay is useful to study the mechanisms of dystrophic endball formation.

PTPσ is expressed on axon terminals. PTPσ-deficient axons can grow on a CSPG-coated substratum, and PTPσ-deficient mice show enhanced axon regeneration after spinal cord injury [36]. The wedge peptide ISP for PTPσ also releases CSPG-mediated inhibition of axon growth and promotes axon regeneration and functional recovery after spinal cord injury [15]. These facts suggest that axon regeneration is inhibited by CS through its interaction with PTPσ. Consistent with this idea, both HS and CS bind to the Ig1 domain of PTPσ (Figure 1). HS oligomerizes, while CS monomerizes, PTPσ [1]. Monomerization of PTPσ by CS activates its tyrosine phosphatase activity [1,36]. Cortactin is then dephosphorylated by PTPσ [37]. Because phospho-cortactin is essential for actin polymerization, which is required for lysosome-autophagosome fusion [38], PTPσ activation disrupts autophagy flux, leading to autophagosome accumulation. These continuous reactions result in dystrophic endball formation and inhibit axon regeneration (Figure 3) [37]. HS exerts an opposite effect on axon regeneration. Thus, chondroitin sulfate *N*-acetylgalactosaminyltransferase-1-deficient mice show a striking axon regeneration after spinal cord injury [39]. As HS and CS share the same tetrasaccharide linkage to the core protein, and this enzyme is responsible for CS synthesis, its ablation results in increased HS synthesis. It is thus likely that the striking axon regeneration is due to a combined effect of increased HS and decreased CS.

Why do CS and HS have the exact opposite effects on PTPσ activation? The CS-E unit, a highly sulfated form of CS, preferentially binds to PTPσ among a variety of sulfation forms [37]. As CS-E accounts for only 3% of CS in the CNS [40], a short stretch of CS-E, such as tetrasaccharide, can only be formed in CS in the CNS, which leads to monomerization and activation of PTPσ. On the other hand, most sulfated HS but not non-sulfated HS binds to PTPσ [37]. Sulfated HS accounts for about 50% of HS in the CNS [41]. Therefore, a relatively long stretch of sulfated HS can be formed in the HS in the CNS, and this stretch of sulfated HS oligomerizes and suppresses PTPσ. This is how PTPσ serves as a molecular switch in combination with HS or CS to promote or inhibit axon regeneration, respectively [37]. Consistent with this idea, enoxaparin can cluster PTPσ, induce dephosphorylation of cortactin and promote functional recovery after spinal cord injury [42]. Enoxaparin is clinically used as an anticoagulant and is a mixture of heparin oligosaccharide (an oversulfated type of HS) derivatives with an average molecular weight of 45 kDa. In contrast, fondaparinex consists of heparin pentasaccharide with a molecular weight of 17 kDa and cannot exert such effects [42]. The size of fondaparinex may be too short to induce oligomerization of PTPσ.

These contrasting functions of HS and CS are also seen in Sema 5A. HS and CS bind to the same domain, the thrombospondin repeats, of Sema 5A [43]. Sema 5A is a transmembrane protein expressed in developing axons of the *fasciculus retroflexus*, a diencephalon tract associated with limbic function. Sema 5A promotes axon extension through interaction with HS on the axons. On the other hand, extracellular CS converts Sema 5A to an inhibitory cue.

LAR is another receptor for CS. CS binds to the Ig1 domain of LAR, which has a structure similar to that of PTPσ [1]. CS enhances the phosphatase activity of LAR expressed in COS-7 cells. LAR inhibitors promote axon regeneration after spinal cord injury [44].

#### 2.2.2. Substrates of Type IIa RPTPs

RPTPs may govern a broad range of substrates to elicit diverse biological activities. A Drosophila homolog of type IIa RPTPs, Dlar, interacts with Abl and Ena, both of which play a role in actin polymerization [45]. PTPσ and LAR interact with p250GAP and Trio, respectively; p250GAP and Trio also regulate actin dynamics [46,47]. In this context, it is of note that the newly identified PTPσ substrate cortactin is also essential for actin polymerization, which in turn is important for autophagosome–lysosome fusion [37].

Furthermore, LAR interacts with β-catenin to promote epithelial cell migration [48]. N-Cadherin is also a substrate for PTPσ in the inhibition of axon growth [49]. Dlar interacts with liprin-α for synaptogenesis [50]. Liprin-α has also been shown to interact with human type IIa RPTPs [51]. Because liprin is a key molecule in the active zone scaffold of synapses, its interaction with type IIa RPTPs is important for presynaptic differentiation.

However, most of the substrates of type IIa RPTPs remain to be identified. BioID is a method to biotinylate interacting proteins of a target protein. Using this method, approximately 100 proteins interacting with PTPσ have recently been identified [52]. These include liprin-α1, Trio, and cortactin. Thus, it is expected that these proteins may include substrates of PTPσ and may provide clues that could help to uncover the landscape of PTPσ functions.

#### 2.2.3. Autophagy in Neurodegenerative Diseases

Autophagy in neuronal axons is initiated at the axon terminal, and autophagosomes are then fused with lysosomes and retrogradely transported along microtubules toward the cell body. When axons are severed, such as by spinal cord injury, axon terminals and shafts develop dystrophic endballs, where autophagosomes accumulate, indicating that autophagy flux is stagnant. Similarly, in many neurodegenerative diseases, autophagosome accumulation is observed in the synaptic region. Autophagy disruption in synapses precedes neuronal cell death. In this way, it can be inferred that autophagy is essential for neural function. Indeed, many neurodegenerative diseases show defects in autophagy. For example, Parkinson’s disease is closely related to autophagy deficiency. Mutations in the PINK1 and Parkin genes cause mitophagy defects [53,54,55], while abnormalities in Huntington’s disease cause failure to initiate autophagy due to Beclin1 defects [56]. Mutations in the autophagy receptors p62 and OPTN1 cause selective autophagy defects in amyotrophic lateral sclerosis (ALS); TBK1 is a kinase responsible for the phosphorylation of OPTN1, and its mutations are associated with ALS and frontotemporal dementia [57]. Although it is not known whether type IIa RPTPs and GAGs are involved in neurodegenerative diseases, their contribution could be an interesting subject to study. In this context, it is worth noting that enhanced CSPG expression is associated with the lesions of Alzheimer’s disease [58] and ALS [59].

#### 2.2.4. Axon Extension vs. Synapse Formation

So far, we have discussed autophagy flux disruption as a pathologic event. However, autophagy flux disruption might also have a physiological role [60]. This is because neurons are surrounded by an extracellular matrix rich in CS, known as perineuronal nets. Therefore, regenerating or developing axons will encounter this matrix. The CS-PTPσ interaction may stop axon growth through cortactin dephosphorylation and autophagy flux disruption, as described above. This axonal arrest may be important for synapse formation with an appropriate target neuron. In addition, as cortactin is regarded as an essential regulator of protease secretion [61], the cortactin dephosphorylation by PTPσ may also reduce the secretion of proteases that degrade extracellular matrix proteins for axon growth. Consistent with this, the PTPσ blocking peptide ISP could enhance secretion of the lysosomal enzyme cathepsin B and promote extracellular CSPG degradation [62]. PTPσ is highly upregulated at axon terminal and adheres to CSPG-substratum in vitro [63,64]. Dystrophic endballs form stable bonds with oligodendrocyte progenitor cells or pericytes producing CSPGs in the lesion core [63,65]. These facts collectively support the hypothesis that regenerating or growing axons are arrested at their target neurons through the CS-PTPσ axis [60] (Figure 4). PTPσ on presynapses is further involved in synapse formation with its partners on the postsynapses; this is described in the next section.

### 2.3. Synaptogenesis

#### 2.3.1. Synapse Organizers

The synapse is an intercellular junction between a presynaptic neuron and a postsynaptic cell (a neuron or muscle cell). It has a specialized structure for fast, specific information transfer via neurotransmitters and their receptors. This structure is organized by trans-synaptic cell-adhesion molecules. Many of the presynaptic adhesion molecules have been identified, i.e., type IIa RPTPs, neurexins, teneurins, FLRTs, netrin G1 and G2, neuronal pentraxins, and SynCAMs [66,67]. Among these molecules, type IIa RPTPs and neurexins have the most postsynaptic partners (Figure 5A) [66,67]. They may determine the identity of synapses and neuronal networks through heterophilic adhesion with their postsynaptic ligands. For example, all type IIa RPTP family members bind to NGL3 to exclusively organize excitatory synapses. PTPδ binds to IL1RAPL1 and IL1RAcp to exclusively organize excitatory synapses, while it also binds to Slitrks to form inhibitory synapses. TrkC is a postsynaptic ligand of PTPσ in excitatory synapses, but it does not bind to either LAR or PTPδ. In this section, we first describe type IIa RPTPs and their ligands and next address neurexins. However, these two classes also engage in cross-talk with one another. This will be discussed in the next section.

Type IIa RPTPs exist on presynapses and serve as synapse organizers with various postsynaptic partners (Figure 5A) [68]. Netrin-G ligand 3 (NGL3) is the first identified postsynaptic ligand for type IIa RPTPs [9,10]. NGL3 forms excitatory synapses and binds to all type IIa RPTPs, i.e., LAR, PTPδ, and PTPσ [10]. NGL3 induces presynaptic differentiation via clustering of vesicular proteins that are specific to excitatory synapses (e.g., vesicular glutamate transporter 1) but not to inhibitory synapses (e.g., vesicular GABA transporter) [9,10]. The binding is mediated through the first leucine-rich repeats (LRR) of NGL3 and the first two FNIII domains of type IIa RPTPs [9,10].

TrkC on postsynapse binds to PTPσ to induce excitatory presynaptic differentiation (Figure 5A) [2]. This binding is specific to PTPσ, and thus TrkC does not bind to LAR or PTPδ. The canonical TrkC ligand neurotorophin-3 (NT-3) derived from presynaptic neurons is essential for TrkC-dependent dendritic growth and morphogenesis of Purkinje neurons [69] and promotes PTPσ- and TrkC-dependent synapse organization of hippocampal neurons [70]. These data suggest that the complex of presynaptic PTPσ, presynaptic neuron-derived NT-3, and postsynaptic TrkC plays a role in excitatory synapse formation. Mini exon insertions at MeA and MeB in the PTPσ Ig domains produced by alternative splicing influence the binding between TrkC and PTPσ, with MeA (−) MeB (−) having the strongest impact [5].

IL1RAPL1 [3,4] and IL1RAcp [5] have LRRs and are postsynaptic ligands for PTPδ that organize excitatory synapses (Figure 5A). This is in sharp contrast to PTPδ-organizing inhibitory synapses with Slitrks, which is described below. PTPδ-deficient neurons lose excitatory synapse activity [3]. IL1RAPL1-deficient mice exhibit a decrease in dendritic spine density and deficits in learning [71], which is consistent with the association of IL1RAPL1 mutations and X-linked mental retardation and startle epilepsy [72,73]. Alternative splicing in the Ig domains of PTPδ also influences the binding to IL1RAPL1 and IL1RAcp [3,4], with MeA (+) MeB (+) having the strongest effect [68].

Slitrk1-6 are LRR-containing transmembrane proteins. Slitrk3 binds to PTPδ to induce inhibitory synapse differentiation (Figure 5A) [74]. Other Slitrks also bind to PTPδ for inhibitory synapse formation and PTPσ for excitatory synapse formation (Figure 5A) [75]. The N-terminal LRR of Slitrks and three Ig domains of PTPδ or PTPσ are required for the binding [6]. An insertion at MeB in PTPδ or PTPσ is essential for the binding. A subset of Slitrks have been linked to schizophrenia, autism, Tourette syndrome, and obsessive-compulsive disorder.

All SALMs have LRR domains, but only SALM3 and -5 are involved in synaptogenesis for excitatory and inhibitory synapses (Figure 5A) [76]. SALM3 binds to all type IIa RPTPs for excitatory synapse formation, where the MeB insert is required [7].

It has recently been elucidated that PTPσ and PTPδ interact with collagen XXV, and this interaction is indispensable for intramuscular motor innervation [77]. Mutations in COL25A1, which encodes collagen XXV, are associated with a congenital cranial dysinnervation disorder.

An important question is whether the phosphatase activity of type IIa RPTPs is required for the presynaptic differentiation. The intracellular D2 domain of PTPσ is essential for synapse formation by Slitrk1 or TrkC, as the D2-deficient mutant cannot induce synapses with these postsynaptic partners [78]. Double knockdown of the D2-interacting proteins liprin-α2 and -α3 cannot induce synapse formation by PTPσ and Slitrk6. Moreover, knockdown of the PTPσ substrate N-cadherin or p250RhoGAP cannot form synapses by PTPσ and Slitrk6 [78]. These facts collectively suggest that the intracellular domains of type IIa RPTPs are indispensable for synapse induction, but the importance of its enzymatic activity remains elusive.

In addition to type IIa RPTPs, neurexins are another major presynaptic adhesion molecule. Neurexins bind to a variety of postsynaptic adhesion molecules. Neuroligins, LRRTMs, dystroglycan, GABA-A receptors, latrophilins, and calsyntenins are representative postsynaptic partners of neurexins (Figure 5A). Neurexins also bind to protein complexes, such as BAI3/C1qls, GluK/C1qls, and GluD1,2/Cbln1-3 (Figure 5A). Mutations of neurexins and of their partners (neuroligins, LRRTMs, dystroglycan, GABA-A receptors, latrophilins, GluK, and GluD1) are associated with neuropsychiatric disorders, indicating the physiological significance of synapses involving neurexins [66,67].

#### 2.3.2. HS Involved in Synapse Formation

The minimum structures required for the binding between TrkC and PTPσ are TrkC LRR-Ig1 domains and PTPσ Ig1–3 domains. Electrostatic interactions are indispensable; these involve D240 and D242 in the first Ig domain of TrkC and R96 and R99 in the first Ig domain of PTPσ [79]. The residues of PTPσ form part of the extended positively charged surface on PTPσ Ig1 domain that is required for the binding with HS and CS [1]. Indeed, TrkC and HSPGs compete for PTPσ binding [79]. The binding between PTPσ and HSPGs is thought to be cis-binding, while that of PTPσ and TrkC is trans-binding. It has been proposed that the shift from cis-binding to trans-binding would explain how axon growth shifts to synapse formation [79]. Combined with the discussion in Section 2.2.4, this could be further extended to a hypothesis that axons regenerate or grow with the aid of cis-binding of HS and PTPσ, arrest at their target neurons through the axis of CS-PTPσ, and consequently organize synapses via trans-binding of PTPσ and its postsynaptic ligands, such as TrkC (Figure 4).

GPC4 is an HSPG essential for synapse formation. Proteolytically cleaved GPC4 (probably by a furin-like convertase) binds to presynaptic PTPσ and postsynaptic LRRTM4 to organize excitatory synapses (Figure 5B) [8]. HS in the GPC4 moiety is indispensable for these bindings, since HS deletion by a mutation at the HS linkage site in GPC4 or GPC4 knockdown reduces the complex formation. In addition, a mutation at the HS-binding site in PTPσ cannot rescue the effect of PTPσ knockdown [8]. GPC4 binds to PTPδ in excitatory synapse formation [80]. GPC4 and -6 were identified among astrocyte secreted factors to promote the formation of synapses of retinal ganglion cells via GluA1 AMPA receptors [81]. Although the postsynaptic ligand for the PTPδ-GPC4 complex remains to be identified, this interaction induces neuronal pentraxin 1 (NP-1) release from the presynapse and consequently cluster postsynaptic AMPA receptors (Figure 5B) [80]. These findings highlight the importance of crosstalk between neurons and astrocytes in synapse organization. The interaction between type IIa RPTP and GPC is also found in Drosophila neuromuscular synapses. Thus, neuronal terminal Dlar (a type IIa RPTP homolog) binds to muscular Dally-like (a GPC4/6 homolog) to form neuromuscular junctions [82]. Dlar also forms a complex with another HSPG syndecan [82,83,84].

It has recently been established that neurexins are HSPGs [85]. The binding between neurexins and their postsynaptic ligands, such as neuroligins and LRRTMs, requires not only protein–protein interaction but also HS–protein interaction (Figure 5B). Mice lacking HS on neurexin 1 show structural and functional deficits in synapses in the CNS [85]. PTPσ and neurexin bind one another on presynapse, and further bind to postsynaptic LRRTM3 and -4. HS in the neurexin moiety is essential for the binding between these three molecules (Figure 5B) [86].

As described above, mutations in neurexins and their postsynaptic ligands have been linked with neuropsychiatric disorders. In this context, it is of note that HS deficiency caused by EXT1 or HS3ST5, which encodes an HS synthetic enzyme or an HS sulfotransferase, respectively, are associated with autism [87,88]. HS expression is decreased in the subventricular zone of the lateral ventricles of autism postmortem brains [89]. In addition, postnatal inactivation of Ext1 in mice leads to an autism-like phenotype [90].

## 3. Perspective

The HSPG GPC4 links presynaptic PTPσ and postsynaptic LRRTM4. The presynaptic HSPG neurexins also link PTPσ and LRRTM3 or -4. HS in the HPSG moiety is indispensable for these bindings. Moreover, among a variety of postsynaptic ligands, at least neuroligins and LRRTM3 and -4 bind to neurexins in an HS-dependent manner. We may understand only a small portion of HS-dependent synaptogenesis, and the significance of HS in synaptogenesis will almost certainly be expanded.

Although this review has mainly described the functions of type IIa RPTPs in axon regeneration and synapse formation, there are other functions for these molecules. As mentioned above, LAR interacts with β-catenin to promote epithelial cell migration [49]. PTPσ inhibits class II PI3K (Vps34) and autophagy [91]. PTPσ allosteric inhibitor DJ001 can inhibit hematopoietic stem cell regeneration [92]. Comprehensive identification of substrates of type IIa RPTPs is useful not only for presynapse differentiation and axon regeneration but also for other pathologic (e.g., cancer) and physiologic events. We have not yet glimpsed the whole landscape of the functions of type IIa RPTPs.

## Figures and Tables

**Figure 1 ijms-22-05524-f001:**
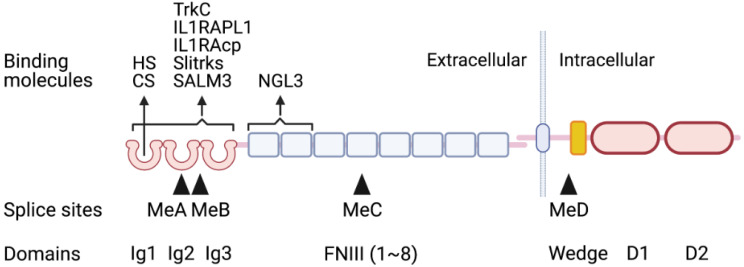
Structure of type IIa RPTPs. Domains and splice sites of type IIa RPTPs are indicated. Binding molecules and their binding domains are also depicted.

**Figure 2 ijms-22-05524-f002:**
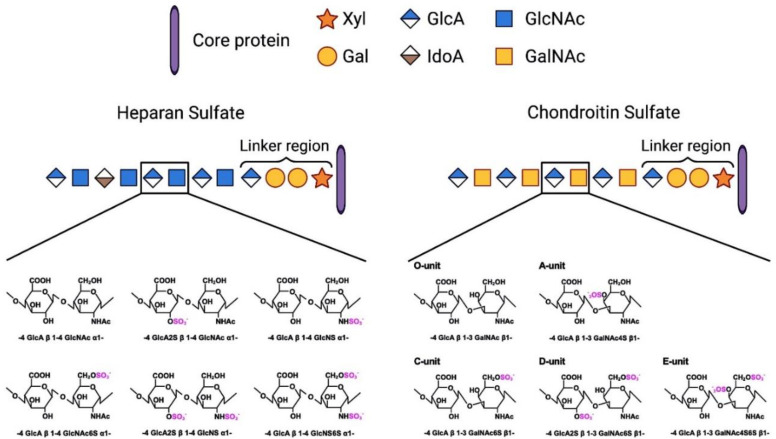
Chemical structures of HS and CS. Among a variety of sulfation patterns, representative ones are depicted. Gal, Galactose; GalNAc, *N*-acetylgalactosamine; GlcA, Glucuronic acid; GlcNAc, *N*-acetylglucosamine; IdoA, Iduronic acid; Xyl, Xylose.

**Figure 3 ijms-22-05524-f003:**
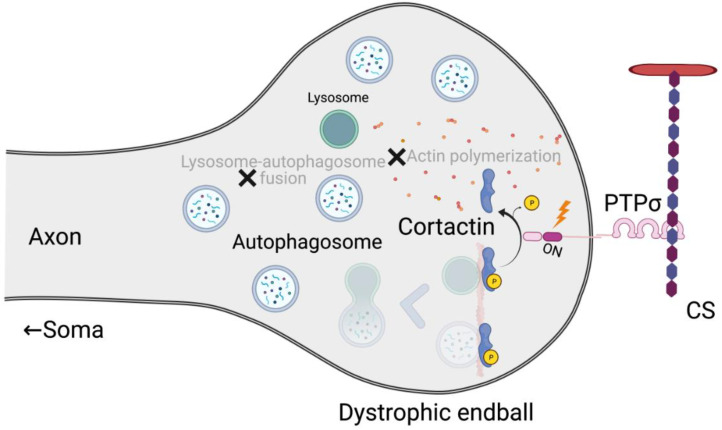
Mechanisms of dystrophic endball formation. Disruption of autophagy flux through the axis of CS and PTPσ is depicted.

**Figure 4 ijms-22-05524-f004:**
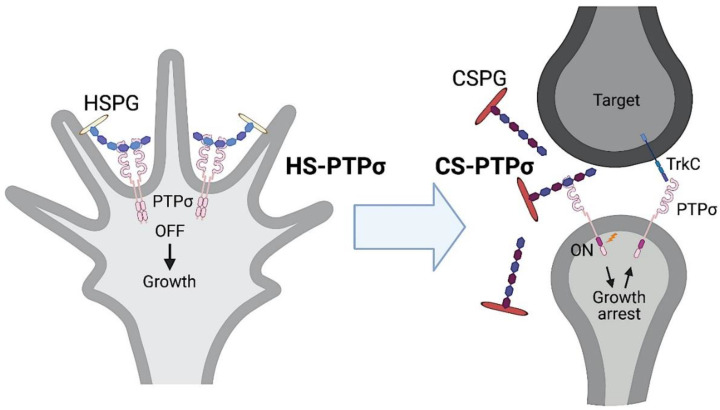
Switch model of axon growth and synapse formation. Axons grow through *cis*-binding of HS and PTPσ, stop through *trans*-binding of CS and PTPσ, and form synapse involving PTPσ and its postsynaptic partners, such as TrkC.

**Figure 5 ijms-22-05524-f005:**
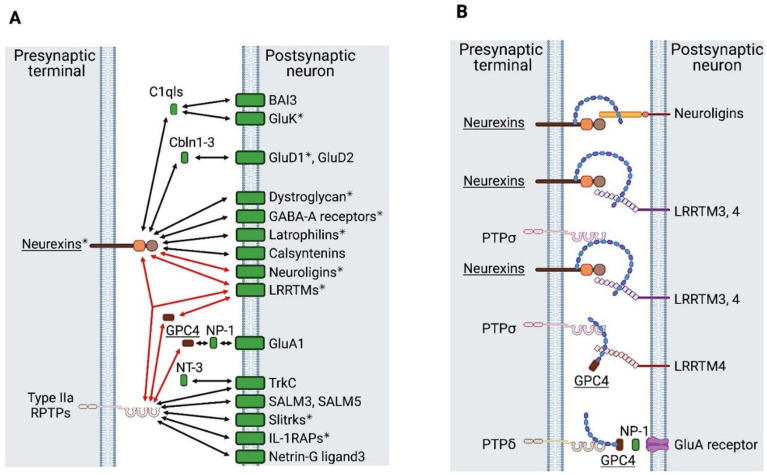
Trans-interaction of synaptic organizers (**A**). Neurexins and type IIa RPTPs are depicted as representative presynaptic adhesion molecules. Red arrows show the interactions where HS is involved. Asterisks indicate genetic association with neuropsychiatric disorders [66]. HSPGs are indicated by underlines (**B**). Trans-interactions of synapse organizers involving HSPGs are shown. HSPGs are indicated by underlines.

## Data Availability

Not applicable.

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
