# Peer review of "Type IIa RPTPs and Glycans: Roles in Axon Regeneration and Synaptogenesis"

_ijms, 2021, doi:10.3390/ijms22115524_

Round 1

Reviewer 1 Report

This review is timely and sound. The only two requirements made by this reviewer is (1) to add a figure containing the chemical structures of the major glycans discussed in the paper and (2) to expand the discussion of contributing factors that downregulate neural regeneration and growth/development such as chondroitin sulfate in certain conditions (doi: 10.4103/1673-5374.206630) and the chondroitin sulfate-rich molecular assembly, aggrecan (doi: 10.1515/revneuro-2020-0146). It is important for the readers to see that neuroplasticity is an event in which molecular players have their ultimate function intrinsically controlled by many spatial and temporal factors. The authors would improve considerably their review paper if more information about how this complex mechanism is regulated by glycans. 

Author Response

Reviewer 1.

This review is timely and sound. The only two requirements made by this reviewer is (1) to add a figure containing the chemical structures of the major glycans discussed in the paper and (2) to expand the discussion of contributing factors that downregulate neural regeneration and growth/development such as chondroitin sulfate in certain conditions (doi: 10.4103/1673-5374.206630) and the chondroitin sulfate-rich molecular assembly, aggrecan (doi: 10.1515/revneuro-2020-0146). It is important for the readers to see that neuroplasticity is an event in which molecular players have their ultimate function intrinsically controlled by many spatial and temporal factors. The authors would improve considerably their review paper if more information about how this complex mechanism is regulated by glycans.

Response:

(1)We added structures of HS and CS as new Figure 2.

(2)We addressed CS functions regarding neural plasticity and neurodegenerative diseases in the section 2.1. We added 9 new reference papers to our list.

Reviewer 2 Report

This is a well written original review describing the functions of type IIa RPTPs in axon regeneration and synapse formation. The manuscript is well documented and suggests models and mechanisms of dystrophic endball formation in injured axon terminals, axon growth and synapse formation, and trans-interaction of synaptic organizers, which might be interesting for neuroscientists.

Author Response

Reviewer 2.

This is a well written original review describing the functions of type IIa RPTPs in axon regeneration and synapse formation. The manuscript is well documented and suggests models and mechanisms of dystrophic endball formation in injured axon terminals, axon growth and synapse formation, and trans-interaction of synaptic organizers, which might be interesting for neuroscientists.

Response: We appreciate the reviewer’s positive evaluation of our manuscript.

Round 2

Reviewer 1 Report

The authors have properly addressed all concerns raised by the reviewer. The revised paper can be now accepted.